# Operator-Discretized Representation for Temporal Neural Networks

## Abstract

This paper proposes a new representation of artificial neural networks to efficiently track their temporal dynamics as sequences of operator-discretized events. Our approach takes advantage of diagrammatic notions in category theory and operator algebra, which are known mathematical frameworks to abstract and discretize high-dimensional quantum systems, and adjusts the state space for classical signal activation in neural systems. The states for nonstationary neural signals are prepared at presynaptic systems with ingress creation operators, and are transformed via synaptic weights to attenuated superpositions. The outcomes at postsynaptic systems are observed as the effects with egress annihilation operators (each adjoint to the corresponding creation operator) for efficient coarse-grained detection. The follow-on signals are generated at neurons via individual activation functions for amplitude and timing. The proposed representation attributes the different generations of neural networks, such as analog neural networks (ANNs) and spiking neural networks (SNNs), to the different choices of operators and signal encoding. As a result, temporally-coded SNNs can be emulated at competitive accuracy and throughput by exploiting proven models and toolchains for ANNs.

## 1 Introduction

Modern neural networks are expected to solve demanding AI problems with datastreams in extremely high dimensions. Under widely-available computing infrastructure, the situation is becoming even more challenging, when the neural dynamics for data processing is inherently temporal and online as in the biological systems [1–4]. An appropriate neural network representation for natively handling sequences of timestamped events should significantly improve computational efficiencies. When event sequences are processed with artificial neural networks, known techniques typically compute layer-wise outputs synchronously at every discretized time step to align their data and computing wavefront, as seen in recent investigations on SNNs [5–7] or time series forecasting [8–11]. Though algorithms may sometimes be given in event-driven manners, their execution in SW has to resort to fine-grained synchronous discretization [12–15] or closed-form approximations of temporal dynamics that require exact temporal ordering of the events [14, 16]. As a result, accuracies competitive to ANNs have only been obtained at an expense of throughput and scalability.

In temporally executing neural networks in commercial systems, the period $T_c$ of the global clock is typically chosen small enough compared with the characteristic time of the neural dynamics $t_0$:

$$T_c \ll t_0, \tag{1}$$

to precisely track the temporal dynamics, for example, the membrane potential changes to determine the next firing timing of SNNs. This is a sharp contrast to how the biological brain operates with low-frequency brain waves [17] closer to our behavioral time scale:

$$T_c \gg t_0. \tag{2}$$

Energy and functional efficiencies can be significantly improved if a new representation can avoid synchronously computing the temporal dynamics at every small time step by better decoupling different time scales, It is tempting for those with some physics background to apply techniques being developed for quantum systems since they are naturally asynchronous events in extremely high dimensions. Indeed, operator algebra has been applied to Hopfield networks [18] as well as other classical systems [19–21]. However, since operators are used for stationary neuron states out of spins and charges rather than those for nonstationary neural signals traveling over axon-synapse-dendrite networks, its full potential has not been extracted for modern temporal workloads.

Here in this paper, we propose a new representation of neural networks that can efficiently compute their dynamics as coarse-grained sequences of operator-discretized events. Our approach takes advantage of existing mathematical frameworks that have been originally developed to abstract and discretize high-dimensional quantum systems. These techniques are, with necessary modifications, applied to neural networks that are also high dimensional, but inherently are classical. Different generations of neural networks, such as ANNs and SNNs, are attributed to different choices of operators and signal encoding. Our formulation can efficiently emulate temporally-coded SNNs with fully exploiting existing assets, such as models and toolchains for ANNs. It should be noted that the scope of this paper is on classical neural networks, though the proposed representation may bring us a new perspective on AI and quantum computing (QC) [22],

## 2 Logical representation

Let us start with the logical aspects. Figure 1 presents diagrammatic representations for quantum and neural networks. In short, once the state spaces are respectively defined, they look surprisingly similar, in particular when we regard qubits as nonstationary and flying [23] as well.

### 2.1 Logical abstraction and state space

The operation of neural networks is to be abstracted by exploiting diagrammatic notions of categorical theories [24–26]. These techniques have been applied both to quantum and classical systems and their processes without much referring to actual physics inside [27]. Here, we will consider pure states only (i.e., wave function vectors rather than density matrices) for quantum, since our purpose is to explicitly compare quantum and classical networks.

A known categorical diagram for a quantum network is examplified in Fig. 1 (a). It consists of three major blocks: the states, the processes/transformations, and the effect, for preparation, operation, and observation of quantum systems, respectively. Without operation, the inner product of the state $|\rho\rangle$ represented by a tensor product of each qubit $|\rho_i\rangle$ state prepared at quantum system $\mathcal{S}_i^Q$

$$|\rho\rangle = |\rho_1\rangle \otimes .... \otimes |\rho_n\rangle. \tag{3}$$

and the effect $\langle\alpha|$ represented by a tensor product of each effect $\langle\alpha_i|$ at quantum system $\mathcal{R}_i^Q$

$$\langle\alpha| = \langle\alpha_1| \otimes .... \otimes \langle\alpha_n|, \tag{4}$$

can compute the conditional probability $P(\alpha|\rho)$ as

$$|\langle\alpha|\rho\rangle|^2 = \prod_{i=1}^{n} |\langle\alpha_i|\rho_i\rangle|^2 = \prod_{i=1}^{n} P(\alpha_i|\rho_i) = P(\alpha|\rho). \tag{5}$$

In general, the probabilities cannot be factorized this way other than for the slices, providing a rich set of non-classical computing power, such as with entanglement, to quantum networks.

The corresponding diagram for a classical neural network is proposed in Fig.1 (b). The states for neural signals are prepared at presynaptic systems. They are transformed into weighted sums via synaptic networks. The outcomes are observed at postsynaptic systems as the effects to generate the follow-on states and signals. As is the quantum case, we assume that the transformations in axon-synapse-dendrite networks are linear. We define, in analogy to the qubit, the cubit, which stands for the abbreviation of *classical universal bit*, for neural signals. Though the definition is informational, rather than physical, we inherit Dirac notation but with double bras and kets, indicating that the

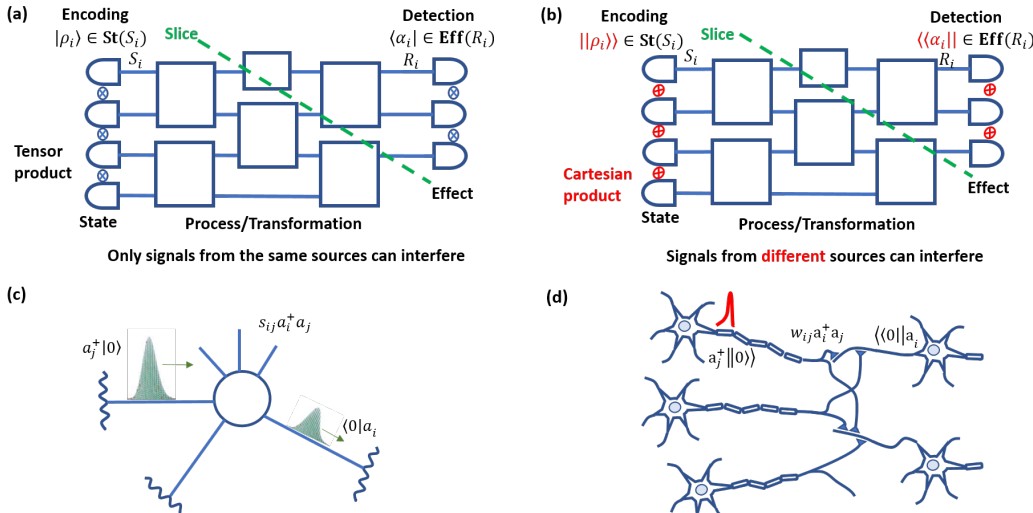

Figure 1: Diagramatic comparison of quantum and neural networks: (a) Quantum network consisting of states, processes/transformations, and effects; (b) Corresponding diagram for a neural network; (c) Operator representation for creation, scattering, and annihilation of quantum wave packets; (d) Operator representation for creation, weighted sum, and annihilation of neural signals. Note that weight matrix $w_{ij}$ in (d) corresponds to scattering matrix $s_{ij}$ in (c).

states consist of macroscopic ensembles of qubits [1]. Multiple types of logical cubits are defined:

| | | |
|---|---|---|
| **Normalized full cubit** | $\|\|c\rangle\rangle := \bar{c}\,\|\|0\rangle\rangle + c\,\|\|1\rangle\rangle, \|\bar{c}\|^2 + \|c\|^2 = 1$ | $\in U(1)$ or $SO(2)$ |
| **Normalized half cubit** | $\|\|c\rangle\rangle := c\,\|\|1\rangle\rangle, 0 \leq \|c\|^2 \leq 1$ | $\in U(1) \cap \mathrm{R}$ |
| **Unnormalized full cubit** | $\|\|c\rangle\rangle := \bar{c}\,\|\|0\rangle\rangle + c\,\|\|1\rangle\rangle$ | $\in \mathrm{R}^2$ or C |
| **Unnormalized half cubit**. | $\|\|c\rangle\rangle := c\,\|\|1\rangle\rangle$ | $\in \mathrm{R}$ |

$$(6)$$

The information encoded to cubits is assumed to be real for simplicity but can be complex for complex-valued neural networks [28].

A set of cubits $\|\|\rho\rangle\rangle$ can compactly be represented by Cartesian product (or coproduct in category theory terminology) of each cubit $\|\|\rho_i\rangle\rangle$ at axon $S_i^C$ as

$$\|\|\rho\rangle\rangle = \|\|\rho_1\rangle\rangle \oplus .... \oplus \|\|\rho_n\rangle\rangle . \tag{7}$$

They are to be detected by effect $\langle\langle\alpha\|$ consisting of $\langle\langle\alpha_i\|$ via dendrite $R_i^C$ as:

$$\langle\langle\alpha\| = \langle\langle\alpha_1\| \oplus .... \oplus \langle\langle\alpha_n\| . \tag{8}$$

Based on an argument for the linear systems in [29], the norm $p$ for cubits is expected to be either1 or 2, Euclidean norm ($p = 2$), which is also found in wireless communication and signal processing literature [30] (e.g., $\|\|1\rangle\rangle$ and $\|\|0\rangle\rangle$ for I and Q), makes sense to represent wave-like dynamics [31–34] in complex-valued state spaces, while Manhatten norm ($p = 1$) is for ordinary real-valued state spaces typically assumed for classical probabilistic computing [29]. Under the linear weighted sum transformations in Cartesian-product state spaces, the log encoding [35] can consistently relate the summation of the inner product for each cubit to the mutiplication of the corresponding probabilities for the product event via bias thresholds $P_i$'s and $P_{total} = \prod_{i=1}^{n} P_i$ as

$$|\langle\langle\alpha\|\|\rho\rangle\rangle|^p = \sum_{i=1}^{n} |\langle\langle\alpha_i\|\|\rho_i\rangle\rangle|^p \sim \sum_{i=1}^{n} \log \frac{P(\alpha_i|\rho_i)}{P_i} = \log \prod_{i=1}^{n} \frac{P(\alpha_i|\rho_i)}{P_i} = \log \frac{P(\alpha|\rho)}{P_{total}}. \tag{9}$$

## 2.2 Operators as neural computing primitives

Operator algebra is a well-established technique to systematically compute quantum physics problems in high-dimensional tensor-product spaces (or Fock for indistinguishable particles). Interac-

---

[1]Further investigation on the relation between qubits and cubits from a physics point of view is desired.

tions between states are represented by scattering matrices (S-matrices) [36] as exemplified in Fig. 1 (c), Here, we develop an operator formalism in Cartesian-product state spaces for neural networks in Fig. 1 (d).

A neural signal at $S_i^C$ is selectively activated in the entire state space spanned as,

$$||0\rangle\rangle = ||0_1\rangle\rangle \oplus ... \oplus ||0_n\rangle\rangle \text{ and } ||1_i\rangle\rangle = ||0_1\rangle\rangle \oplus ... \oplus ||1_i\rangle\rangle \oplus ... \oplus ||0_n\rangle\rangle. \tag{10}$$

States for concurrently activating multiple neural signals can be given, by specifically noting the activated systems $i$ and $j$ as

$$||1_{ij}\rangle\rangle = ||0_1\rangle\rangle \oplus ... \oplus ||1_i\rangle\rangle \oplus ... \oplus ||1_j\rangle\rangle \oplus ... \oplus ||0_n\rangle\rangle. \tag{11}$$

Thus, $||1_i\rangle\rangle$ can mean a single cubit state for $S_i^C$ only or a multiple cubit state in which only $S_i^C$ is fully activated, depending on the context.

The mutually-adjoint creation and annihilation operators on these states, a and $a^\dagger$ are defined as

$$||1_i\rangle\rangle = a_i^\dagger ||0\rangle\rangle \text{ and } ||0\rangle\rangle = a_i ||1_i\rangle\rangle. \tag{12}$$

Multiple signals can be activated in different systems, for example, by

$$||1_{ij}\rangle\rangle = a_i^\dagger a_j^\dagger ||0\rangle\rangle. \tag{13}$$

Depending on whether $i = j$ is allowed in each $T_c$ or not, they are superficially treated like Bosons for rate-coded SNNs (rSNNs) or like Fermions for temporally-coded SNNs (tSNNs).

The transformation $\mathcal{T}_{ij}$ from sender system $\mathcal{S}_j$ to receiver system $\mathcal{R}_i$ is described as:

$$\mathcal{T}_{ij} = w_{ij} a_i^\dagger a_j. \tag{14}$$

Noted that $w_{ij}$ works as the scattering matrix. Cartesian product state space, rather than tensor-product, can incorporate the weighted sum naturally as the superposition of incoming neural signals from different sources. Higher-order interactions are possible, for example as,

$$\mathcal{T}_{ij} = w_{ij} \breve{a}_i^\dagger \hat{a}_i \breve{a}_j^\dagger \hat{a}_j. \tag{15}$$

However, in that case our original assumption of linear synaptic networks is not valid anymore.

The logical neuron model in the operator representation is defined as effects for detecting incoming fragment of signal energies from presynaptic neurons to generate states for the follow-on neural signals. The signal detection process corresponds to the projective measurement in QC, leading to more advanced detection strategies than simple threshold detection strategies. When the fully activated state $||\rho_j\rangle\rangle = a_j^\dagger ||0\rangle\rangle$ is detected by the effect $\langle\langle\alpha_i|| = \langle\langle 0|| a_i$ via $\mathcal{T}_{ij}$,

$$|\langle\langle\alpha_i|| \mathcal{T}_{ij} ||\rho_j\rangle\rangle|^p = |\langle\langle 0| |b_i (w_{ij} b_i^\dagger a_j) a_j^\dagger |0\rangle\rangle|^p = |w_{ij}|^p = \log P(\alpha_i|\rho_j). \tag{16}$$

Nonlinear binary operations such as AND/OR are possible using appropriate activation functions with different thresholds, as those in perceptrons [37].

## 3 Physical representation

The proposed physical representation of neural networks is outlined in Fig. 2. It introduces explicit temporal dependences for operators and neural signals The operators for ingress and egress paths create and annihilate nonstationary neural signals over elastic physical media, i.e., axons ($S_i^C$'s) and dendrites ($\mathcal{R}_i^C$'s).

### 3.1 Operators for eigenmodes

First, the physical representation of the creation and annihilation operators for stationary neural signals $a_i^\dagger$ and $a_i$ are constructed in accordance with the quantum creation and annihilation operators $a_i^\dagger$ and $a_i$ in the one-dimensional transmission line (TL) model in circuit QED [38]. Circuit QED is one of the established baseline theories in QC, which bridges classical circuit dynamics and quantum. The Hamiltonian $\mathcal{H}_{ij}$ for a TL creating consisting of $N$ identical capacitors of the capacitance $C_0$

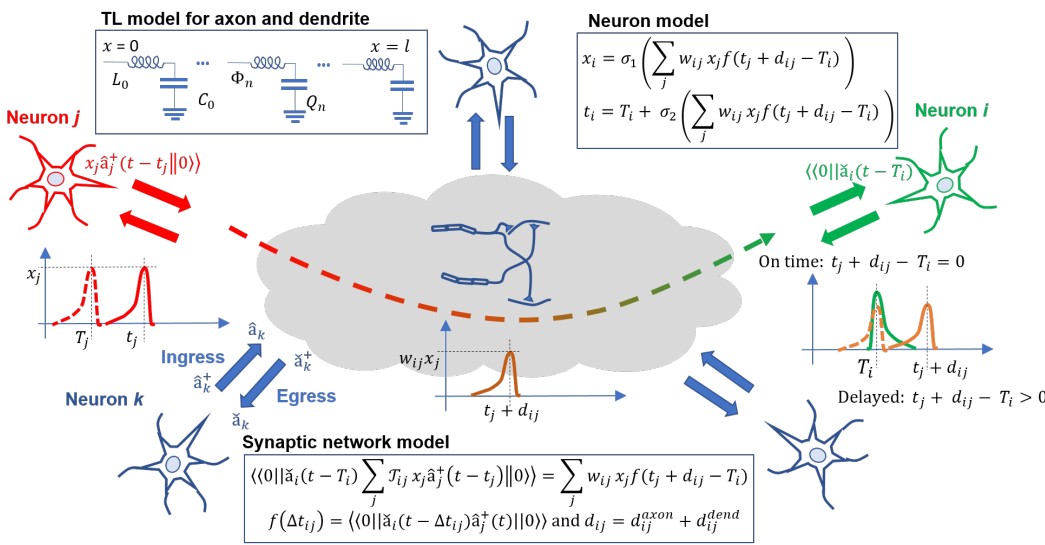

Figure 2: Physical representation of operator-discretized neural networks with explicit local time $t$ dependency with respect to global time $T$. The creation and annihilation operators for ingress and egress paths represent nonstationary neural signal dynamics across axon-synapse-dendrite networks. LC TL models are used for axons and dendrites instead of RC cable models. The neuron model consists of different activation functions for signal amplitude and timing,

(each containing the charge $Q_n$) and $N$ identical inductors of the inductance $L_0$ (each containing the flux $\Phi_n - \Phi_{n-1}$), is given by

$$\mathcal{H}_i = \sum_n \left[ \frac{1}{2C_0}Q_n^2 + \frac{1}{2L_0}(\Phi_n - \Phi_{n-1})^2 \right] = \sum_m \hbar\omega_m a_i^\dagger(k_m, \omega_m)a_i(k_m, \omega_m), \qquad (17)$$

where $m$ is the eigen mode index for a given boundary condition. The lossless LC-based model can better transmit energy and information than the dissipative RC-based biological cable model [39]

We define $\mathrm{a}_i^\dagger(k,\omega)$ and $\mathrm{a}_i(k,\omega)$ as the classical counterpart of $a_i^\dagger(k,\omega)$ and $a_i(k,\omega)$. The following simple linear dispersion for a constant velocity $v$ are assumed in the range of interest:

$$v = \frac{\partial\omega_m}{\partial k_m} = \frac{\omega_m}{k_m} = \text{const.} \quad \forall m. \qquad (18)$$

Consequently, $\mathrm{a}_i^\dagger(k.\omega) = \mathrm{a}_i^\dagger(\omega), \ \mathrm{a}_i(k,\omega) = \mathrm{a}_i(\omega)$. Note that $v$ for neural signals is much slower than $v$ for electrical signals in ordinary TL's [31, 32]. Though our focus is on artificial neural networks, biological implications of the present approach will be further discussed in Appendix.

## 3.2 Operators for nonstationary neural signals

Second, the operators basis is changed from $(k, w)$ to $(x, t)$. For ingress signals

$$\hat{\mathrm{a}}_i^\dagger(x,t) = \sum_m \mathrm{a}_i^\dagger(k_m, \omega_m)\mathrm{A}^*(k_m, \omega_m)e^{-i(k_m x - \omega_m t)},$$
$$\hat{\mathrm{a}}_i(x,t) = \sum_m \mathrm{a}_i(k_m, \omega_m)\mathrm{A}(k_m, \omega_m)e^{i(k_m x - \omega_m t)}. \qquad (19)$$

For egress signals

$$\breve{\mathrm{a}}_i^\dagger(x,t) = \sum_m \mathrm{a}_i^\dagger(k_m, \omega_m)\mathrm{A}^*(k_m, \omega_m)e^{i(k_m x - \omega_m t)},$$
$$\breve{\mathrm{a}}_i(x,t) = \sum_m \mathrm{a}_i(k_m, \omega_m)\mathrm{A}(k_m, \omega_m)e^{-i(k_m x - \omega_m t)}. \qquad (20)$$

142 They represent creation and annihilation of neural signals centered at $x = 0$, and $t = 0$, and sent
143 or received at neuron $i$. To be more specific, for example, a neural signal moving out of neuron $i$
144 created at the start of the TL of a length $l$ is given as

$$\hat{a}_i^\dagger(t) \, ||0\rangle\rangle = \hat{a}_i^\dagger(0, t) \, ||0\rangle\rangle. \tag{21}$$

145 It annihilates at the end of the TL after the geometrically-defined delay $d = l/v < T_c$ as

$$\hat{a}_i(t - d)\hat{a}_i^\dagger(t) \, ||0\rangle\rangle = \hat{a}_i(l, t - d)\hat{a}_i^\dagger(0, t) \, ||0\rangle\rangle. \tag{22}$$

## 3.3 Incorporating physical interaction at synapses

147 When multiple neurons are interconnected via synaptic networks, physical interactions with explicit
148 temporal dependences should be incorporated in addition to the free dynamics described above. We
149 consider here primarily $\mathcal{T}_{ij}$ one-body potential scattering via an elastic scattering center as

$$\mathcal{T}_{ij} = w_{ij}\breve{a}_i^\dagger(t - T_i + d_{ij}^{dend})\hat{a}_j(t - t_j - d_{ij}^{axon}), \tag{23}$$

150 where $d_{ij}^{axon}$ and $d_{ij}^{dend}$ are the delays in axon and dendrite between neurons $i$ ande $j$, respectively.

## 3.4 Neuron model with activation functions for amplitude and timing

152 The proposed representation of neural networks allows for more advanced detection strategies than
153 threshold detection, for example, in LIF neurons usually found in the literature [39] s. This is
154 somewhat inspired by the advancement in detection strategies in communication or storage channels
155 [40]. Let us first consider a simple case when a half-cubit neural signal of the peak amplitude $x_j$
156 from a presynaptic neuron $j$ is generated at $t = t_j$ by applying a creation operator as

$$||\rho_j(t)\rangle\rangle = x_j\hat{a}_j^\dagger(t - t_j) \, ||0\rangle\rangle, \tag{24}$$

157 and observed by a postsynaptic neuron $i$ at $T_i$ directly without a synapse.

$$\langle\langle\alpha_i(t)|| = \langle\langle0|| \, \breve{a}_i(t - T_i), \tag{25}$$

158 In general, the state preparation $||\rho_j(t)\rangle\rangle$ at $t_j$ and the observation $\langle\langle\alpha_i(t)||$ at $T_i$ are not temporally
159 alighed, so by using ingress-egress correlation function $f(\Delta t_{ij}) := \langle\langle0|| \, \breve{a}_i(t - \Delta t)\hat{a}_j^\dagger(t) \, ||0\rangle\rangle$,

$$\langle\langle\alpha_i(t)||\rho_j(t)\rangle\rangle = \langle\langle0|| \, \breve{a}_i(t - T_i)x_j\hat{a}_j^\dagger(t - t_j) \, ||0\rangle\rangle = x_j f(t_j + d_{ij} - T_i) \tag{26}$$

160 for $t_j + d_{ij} - T_i \geq 0$, where $d_{ij} = d_{ij}^{axon} + d_{ij}^{dend}$. We should note that for $\Delta t_1 = \Delta t_2 + \Delta t_3$

$$f(\Delta t_1) = f(\Delta t_2)f(\Delta t_3), \quad f(0) = 1. \tag{27}$$

161 With interactions at synapses, the state preparation and observation between neurons pair $i$ and $j$
162 provides

$$\langle\langle\alpha_i(t)|| \, \mathcal{T}_{ij} \, ||\rho_j(t)\rangle\rangle = \langle\langle0|| \, \breve{a}_i(t - T_i)\mathcal{T}_{ij}x_j\hat{a}_j^\dagger(t - t_j) \, ||0\rangle\rangle = w_{ij}x_j f(t_j + d_{ij} - T_i). \tag{28}$$

163 Thus, the aggregated signal detected at neuron $i$ is

$$\sum_j \langle\langle\alpha_i(t)|| \, \mathcal{T}_{ij} \, ||\rho_j(t)\rangle\rangle = \sum_j \langle\langle0|| \, \breve{a}_i(t - T_i)\mathcal{T}_{ij}x_j\hat{a}_j^\dagger(t - t_j) \, ||0\rangle\rangle = \sum_j w_{ij}x_j f(t_j + d_{ij} - T_i). \tag{29}$$

164 This inner-product-based detection in neural systems corresponds to the projection measurement
165 in quantum systems and is the key to enable efficient coarse-grained detection without tracking
166 the membrane potential at fine-grained time steps. For a given waveform defined by creation and
167 annihilation operators, $f(\Delta t_{ij})$ can extract temporally-coded information. Alternatively, the right
168 operator pair can be defined to meet a given $f(\Delta t_{ij})$. The latter approach is to be taken when
169 applying the present idea to efficient emulation of temporally-coded SNNs.

170 By using appropriate activation functions $\sigma_1$ and $\sigma_2$ for the amplitute and the event firing time,
171 resepctively, the detected signal can be converted to the follow-on signal in neuron $i$ as

$$x_i = \sigma_1(\sum_j w_{ij}x_j f(t_j + d_{ij} - T_i)), \quad t_i = T_i + \sigma_2(\sum_j w_{ij}x_j f(t_j + d_{ij} - T_i)). \tag{30}$$

172 Various nonlienarities can be incorporated via $\sigma_1$ and $\sigma_2$ if necessary.

## 3.5 Learning algorithms with operators

The weight update $\Delta w_{ij}$ for unsupervised algorithms, such as Hebbian and STDP for SNNs, is asynchronously (i.e., without explicit dependency on $T_i$) related to ingress-ingress correlartion $g$ as

$$
\begin{aligned}
\Delta w_{ij} &\sim \langle\langle 0 || \, \hat{\mathrm{a}}_i(t - t_i) \hat{\mathrm{a}}_j^\dagger(t - t_j - d_{ij}) \, || 0 \rangle\rangle \\
&= \langle\langle 0 || \, \hat{\mathrm{a}}_i(t - t_i) \breve{\mathrm{a}}_i^\dagger(t - T_i) \breve{\mathrm{a}}_i(t - T_i) \hat{\mathrm{a}}_j^\dagger(t - t_j - d_{ij}) \, || 0 \rangle\rangle \\
&= g(t_i - T_i) g(T_i - t_j - d_{ij}) = g(t_i - t_j - d_{ij}).
\end{aligned}
\tag{31}
$$

Even and odd functions are chosen for Hebbian and STDP, respectively.

The proposed representation can support various supervised learning algorithms and toolchains, when temporal dynamics is synchronously regulated by a coarse-grain global clock in $n$ cycles as

$$
T_i^{(n)} = nT_c \quad \forall i.
\tag{32}
$$

Fine-grained temporal correlations, such as coincidence, can be passed on to the operator correlations by defining a new global variable $X_i^{(n)} = x_i^{(n)} f(t_i^{(n)})$. Then

$$
x_i^{(n+1)} = \sigma_1\Big(\sum_j w_{ij} X_j^{(n)}\Big), \quad t_i^{(n+1)} = T_i^{(n+1)} + \sigma_2\Big(\sum_j w_{ij} X_j^{(n)}\Big).
\tag{33}
$$

The backward calculation can be performed by using the following relation:

$$
\frac{\partial X_i^{(n+1)}}{\partial X_j^{(n)}} = \frac{\partial X_i^{(n+1)}}{\partial x_i^{(n+1)}} \frac{\partial x_i^{(n+1)}}{\partial X_j^{(n)}} + \frac{\partial X_i^{(n+1)}}{\partial t_i^{(n+1)}} \frac{\partial t_i^{(n+1)}}{\partial X_j^{(n)}} = (f(t_i^{(n+1)}\sigma_1' + x_i^{(n+1)} f' \sigma_2') w_{ij}
\tag{34}
$$

Let us go through how this works further with a specific example in the next section.

# 4 Application to temporally-coded SNN

The relation between ANNs and rate-coded SNNs (rSNNs) has been known [41]. Here, we first theoretically prove that under the proposed representation, temporally-coded SNNs (tSNNs) can be equivalently transformed into ANNs by appropriately assigning the operator via $f$ and encoding via $\sigma_1$ and $\sigma_2$, Then we demonstrate practical benefits of doing so by running some benchmarks.

## 4.1 New perspective on ANN-SNN equivalence

**Proposition 1:** When driven by a global clock of $T_i^{(n)} = nT_c$, operator-descritized neural networks defined by Eqs. 28 and 30 for the neural events $(x_i, t_i)$ with the followng setting consistute ANNs.

$$
\text{ANN: } \sigma_1(x) = *, \; \sigma_2 = 0, \text{ and } f(x) = 1.
\tag{35}
$$

The neural signals stay constant at $X_i^{(n)} = x_i^{(n)}$ for $T_i^{(n)} = nT_c$. The operators become arbitrarily picked single-mode $(k, \omega)$ ones. Perceptrons are constructed with binary inputs and Heaviside step function for $\sigma_1$.

**Proposition 2:** When driven by a global clock of $T_i^{(n)} = nT_c$, operator-descritized neural networks defined by Eqs. 28 and 30 for the neural events $(x_i, t_i)$ with the following setting constitute tSNNs.

$$
\text{tSNN: } \sigma_1(x) = 1 \text{ and } \sigma_2(x) = *.
\tag{36}
$$

The tSNN signals for $X_i^{(n)} = f(t_i^{(n)} - T_i^{(n)})$ take specific spike waveforms defined by nonstationary operators which spread into multiple modes in the $(k, \omega)$ basis. The cut-off $X_{min}$ is defined as

$$
T_j^{(n)} \le t_j^{(n)} \le T_j^{(n)} + T_c \; \Leftrightarrow \; 1 \ge X_j^{(n)} \ge X_{min} = f(T_c).
\tag{37}
$$

**Theorem 1:** tSNN in Proposition 2 with $f'(x)\sigma_2'(x) = 1$ runs equivalently in forward and backward to ANN in Proposition 1 with $\sigma_1(x) = x \cdot (x > X_{min})$ for $X_{min} = f(T_c) > 0$.

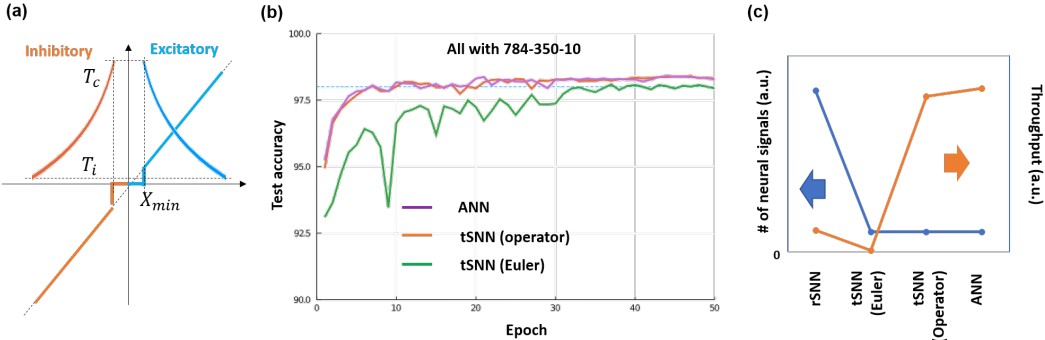

Figure 3: (a) Activation function for operator-discretized tSNNs with excitatory and inhibitory neurons; (b) MNIST benchmark results for ANN, operator-discretized tSNN, and Euler-discretized tSNN; (c) Realtive comparison of the number of neural signals and throughput.

*Proof.* In forward, weighted sum of tSNN reduces to that of ANN as

$$(x_j)_{ANN} = (f(t_j^{(n)} - T_j))_{SNN}, \quad \text{and} \quad (w_{ij})_{ANN} = (w_{ij}f(-T_c + d_{ij}))_{tSNN}. \tag{38}$$

This is because for tSNN,

$$\sum_j w_{ij}f(t_j^{(n)} + d_{ij} - T_i^{(n+1)}) = \sum_j w_{ij}f(-T_c + d_{ij})f(t_j^{(n)} - T_j^{(n)}) = \sum_j w_{ij}f(-T_c + d_{ij}f(t_j^{(n)} - T_i). \tag{39}$$

In backward,

$$\left(\frac{\partial X_i^{(n+1)}}{\partial X_j^{(n)}}\right)_{ANN} = (w_{ij})_{ANN} = (w_{ij}f(-T_c + d_{ij}))_{tSNN} = \left(\frac{\partial X_i^{(n+1)}}{\partial X_j^{(n)}}f(-T_c + d_{ij})\right)_{tSNN}. \tag{40}$$

$\square$

Thus we can emulate tSNN using ANN by renormalizing $w_{ij}$ with the constant $f(-T_c + d_{ij})$.

**Example 1:** We can set tSNN as

$$f(x) = \beta^{-x}, \quad \sigma_2(x) = -\log_\beta x \quad \text{and} \quad T_c = d_{ij} \text{ (i.e.,} f(-T_c + d_{ij}) = 1) \tag{41}$$

$\beta$ works as a base constant to carry or borrow across a fine-grained unit time interval. The logarithmic conversion works as a ReLU activation function in ANN since the conversion is only valid for $X_{min} > 0$. Bipolar neural signals are represented by combining excitatory and inhibitory neurons as shown in Fig.3(a). This setting can also support rSNNs by allowing multiple spikes within $T_c$.

Building blocks in modern ANN models, such as convolution, max pooling, and batch normalization, have to be translated to those in SNNs. The translation is straightforward as long as they are linear transformations. However, batch normalization blocks may require some attention, since they involve nonlinear operations to control both the number and the delay distribution of neural signals.

Once the translations of building blocks are completed, the proposed representation for SNNs can support not only specific models and learning algorithms but a wide variety of them. Under the operator-discretized representation, the inference paths of SNNs can be translated to those of the corresponding ANNs. Thus the standard autograd learning strategy [42] for ANNs equally works without using costly strategies specific to SNNs. The instability associated with differentiating the spike activation function can be avoided by substituting adjoint computation [43] to the operators rather than using arbitrary surrogate functions [6, 44].

## 4.2   Evaluation

Figure 3(b) compares MNIST benchmark results for ANN, Euler-discretized tSNN, and operator-discretized tSNN. We used a stand-alone computing environment without GPU to minimize undesired throughput variations. The code for ANN straightforwardly follows reference implementations

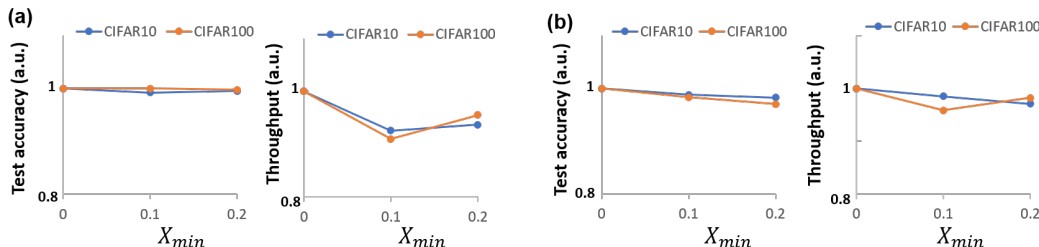

Figure 4: Relative test accuracy and throughput as a function of $X_{min}$ for CIFAR10&100 with resnet18 : (a) With batch normalization; (b) Without batch normalization. $X_{min} = 0$ is for ANN.

and default parameter settings under python 3.8.5 and PyTorch 1.9.1. $lr = 0.001$ with Adam optimizer and is multiplied by 0.9 after every10 epochs. To accommodate Euler-discretized tSNN, simple architecture of 784-350-10 is chosen. The Euler discretization algorithm follows the one in [12], There, forward and backward paths were calculated manually in 30 $\Delta T$ steps in each $T_c$ period. On the other hand,, operator-discretized SNN fully takes advantage of the existing toolchain capabilities of ANN, including autograd. For operator-discretized tSNN, we used the conversion as stated in Example 1 with $X_{min} = 0.1$. In short, the result for operator-discretized tSNN achieves a significantly better throughput, than Euler-discretized one, demonstrating competitive accuracy and throughput to those of ANN.

Figure 3(c) compares the number of neural signals and throughput for rSNN, Euler-discretized tSNN, and operator-discretized tSNN. In the Euler-discretized tSNN, the throughput is severely affected despite the reduction of the number of spikes, Since information is encoded in time rather than in amplitude, naive discretization using fine-grained $\Delta T$ steps is not very efficient in terms of both accuracy and throughput. Indeed, the computing complexity proportionally increases as the number of $\Delta T$ steps, rather than as the number of neural signals. In contrast, both the number of spikes and throughput are comparable to those of ANN in operator-discretized tSNN. The proposed emulation strategy meets computing efficiency without washing out actual neural signal waveforms by embedding fine-grained temporal dynamics into crosscorrelations of operators.

The proposed emulation strategy is expected to be as scalable to larger workloads as ANNs. To validate this assumption, our emulation approach was applied to larger data sets and architectures. Figure 4 summarises the benchmark results for CIFAR10&100 and resnet18. This time, we used SGD with $lr = 0.1$ with batch normalization and $lr = 0.05$ without batch normalization for better convergence. The learning rates were reduced by $\times$ 10 after every 30 epochs for a total of 90 epochs. Again, the ANN code follows reference implementations and default parameter settings in PyTorch documentation, The programs were executed in x86 internal clusters for higher throughput (at an expense of throughput variations due to other jobs) with python 3.6.9 and PyTorch 1.2.0, but again without GPUs. We used multicores in a single node since the conversion between ANN and tSNN is local i.e., not affected by the node configuration. The result confirms that both accuracy and throughput are similarly competitive to ANNs for larger datasets and models. We performed multiple runs for 10 different seeds. The standard deviations were $\lesssim$ 1 % and $\lesssim$ 10 % for accuracy and throughput, respectively.

## 5   Conclusion

This paper proposed a new representation of neural networks that can efficiently compute their dynamics as sequences of operator-discretized events. Our approach takes advantage of existing mathematical frameworks that have been originally developed to abstract and discretize high-dimensional quantum systems with necessary modifications to handle neural networks. Different generations of neural networks, such as ANN and SNN, were attributed to different selections for operators and encoding. Our formulation, when applied to tSNNs, led to a more computationally efficient SW emulation with fully exploiting existing ANN assets. Presently, learning is not perfectly asynchronous because of Eq. 32. However, this limitation makes sense considering that the biological brains also use slow brain waves to efficiently regulate their operations without much affecting online tracking.

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
