# Appendix

## Computing environment

MNIST benchmark runs were executed on a stand-alone x86 PC using 8 CPU cores, 8 GB RAM, and no GPUs, The system installs Windows 10, Jupyter notebook 6.4.3, python 3.8.5, and PyTorch 1.9.1. We used MNIST data in PyTorch divided by a batch size of 128.

CIFAR10&100 benchmark runs were executed on an x86 cluster using 8 cores in a single node, typically 4-5 GB RAM allocated, and no GPUs, The system installs Linux version 8.5. python 3.6.9, and PyTorch 1.2.0. We used CIFAR10&100 data in PyTorch divided by a batch size of 256 [2].

## Present limitations

Currently, the neural network representation is not perfectly asynchronous because of Eq. 32 to pipeline the coarse-grained dynamics. However, this limitation may make sense considering that biological brains also use slow brain waves to lively regulate their operations. The strategy can reduce $w_{ij}$ update frequency without much affecting online tracking performance.

Another limitation is that synaptic networks consisting of the axon, synapse, and dendrite are assumed linear, and nonlinear operations are presently dumped into neurons in conjunction with $\sigma_1$ and $\sigma_2$. For example, batch normalization is a nonlinear operation that may be implemented by more naturally adjusting the amplitude and delay distributions of neural signals. Computing the loss function (in this research, the cross-entropy loss was used) is another case in which the computation is now performed outside operator-discretized networks. Instead of making the synaptic network nonlinear in a Cartesian-product state space, it may stay linear in a tensor product state space, which is not entirely impossible as stated later.

Furthermore, this representation expects the data to take an event-driven (e.g., time-stamped) format, rather than synchronous streams like video data. In latter cases, some sort of front-end to convert frame-synchronous data may help.

## Related topics in AI

If the input data streams are appropriately arranged, the application of the idea to a variety of temporal ANNs beyond tSNNs should be of interest as a follow-on investigation. The operator-discretized representation will make sense for encoding information into temporal sequences and processing them in time as seen in spike trains in biological neural systems. The fine-grained spike dynamics can be nicely decoupled from the coarse-grained behavioral one. We may regard what is happening in the fine-grained part as a sort of vector to time conversion, which is somewhat opposite to time2vec [A1]. This approach will work better when the fine-grained dynamics is dominated by predefined temporal correlations, rather than blindly learned from data.

The operator representation may be combined with other well-established machine learning techniques, such as kernel methods [A2], as a means to constitute appropriate basis sets in large spatiotemporal dimensions. The nonstationary operators can add unique value to such methods by consistently handling temporal dynamics using amplitude and/or temporal coding. Adjoint operators will naturally incorporate backward dynamics for learning. The use of log probability is popular, such as in log-likelihood or Viterbi algorithm, to better deal with product events. Relating the log of the probability to the Euclidean norm of the signal amplitude as in Eq. 9 for $p = 2$ is consistent with, for example, what has been done in Viterbi algorithm under Gaussian noise [A3].

Our emulation strategy can become a searchlight to explore future neuromorphic HW. The bidirectional and elastic nature of our operators may help to natively investigate other physically-oriented (e.g., mechanical) models, such as equilibrium propagation [A4]. The faster turnaround of the proposed emulation methodology will facilitate detailed comparison across multiple design choices, for example, digital and analog spiking [A5, A6] against reduced precision approaches [A7] using modern AI workloads. We believe that temporal coding is essential for neuromorphic HW to be truly as efficient as the biological brain. Encoding information into the pulse width may rather be considered as non-return-zero rSNNs without much information temporarily.

---

[2]Similar results with imagenet in multinode distributed data-parallel to be published upon approval

## Biological implications

Though the present research is on artificial neural networks, originally initiated by treating neural signals as slowly-traveling elastic waves [31, 32], there are growing pieces of works on the importance of wave dynamics in biological neural networks as well [33, 34]. Thus, it may also be worth investigating more biological neuron models, such as Izhikevich neurons [A8] within our representation. Specific functional forms for $\sigma_1$ and $\sigma_2$ will affect amplifying and/or damping of collective wave dynamics, which may better elucidate what is happening in biological systems.

Since the geometrical size of spike signals is less than typical axon-synapse-dendrite network sizes, it makes sense, also from a biological standpoint, to explicitly deal with the traveling wave dynamics of spike signals. The width in size of a spike signal is $\sim 1$ mm for a velocity $\sim$ of 1 m/s and a width in time of $\sim 1$ ms. Thus, the axon-synapse-dendrite networks need to be treated as distributed entities, rather than lumped. The biological implications of present LC TL models for axons and dendrites should be argued further in comparison to the conventional RC cable models. LC TLs are superior in better transmitting signal energy and information without dissipation. However, since the origin of a reasonable amount of $L$ is still controversial, the use of $L$ in the biology literature is limited [A9] to our knowledge.

Here, we speculate that significant $L$ could be caused by more than 4 orders of magnitude heavier masses of ions than that of electrons ($3.81754 \times 10^{-23}$g for Na$^+$ and $9.1093837 \times 10^{-28}$g for e$^-$). This is because the kinetic inductance $L_K$ due to the elastic inertia without scattering given by the following equation [A10] is also more than 4 orders of magnitude higher:

$$L = L_{EM} + L_K, \;\; L_K = \frac{m}{nq^2}\frac{l}{A} \tag{A1}$$

where $m$ is mass, $n$ density, $q$ charge, $l$ length, $A$ area. For the electric TLs, it is well known that, though microscopic electron motions are diffusive, the coherent electrical signal waves driven by the macroscopic charge density offset is not much affected by them. It should be carefully investigated further whether the same situation holds in more electro-mechanical biological environments with much more complex ion dynamics, and therefore, whether the heavier ion masses can indeed make $L_K$ a dominant component as Eq. A1 indicates.

Ingress and egress operators can naturally represent orthodromic and antidromic spike transmissions [A11]. Thus, our operator formalism may help to systematically model bidirectional spike transmissions in biological systems.

## Perspective on future AI and QC

It is one of the primary agendas in future computing how AI and QC would evolve in parallel with conventional computing systems. The present approach will shed new light on it as "AI $\cup$ QC" arguments, alternative to historical "AI $\cap$ QC" [A12], and facilitate us to unlock unknown mechanisms of the brain. This is because the present idea seems to suggest that classical wave dynamics alone can achieve some limited functionalities of QC by taking advantage of Euclidean-norm computing features that have been considered unique to QC [29].

Figure A1 (a) illustrates delay and sum beamforming with delay precoding. It examplifies how the Euclidean norm can enhance the contrast between desired and undesired signals for better performance under a given energy budget (known as the beamforming gain in wireless literature [30]). Freespace should be replaced with a waveguide network for neural signals and the geometrical size will be orders of magnitude reduced as the signal velocity is reduced from the speed of light ($3.0 \times 10^8$ m / s) to $\sim 1$ m/s [31, 32]. Well-arranged attenuated superpositions of neural signals from preceding neurons can increase desired signal amplitudes with constructive interference, and decrease undesired ones with destructive interference. We still need to carefully work on how to exploit this feature in future computing, but let us discuss some interesting possibilities below.

Grover algorithm is well known as a QC algorithm with quadratic speed-up by using amplitude amplification. Here, we would like to argue that the same algorithm is also possible with cubits as shown in Fig.A1(b). By using $n$ normalized full cubits instead of $m = \log_2 n$ qubits, the state $||\rho\rangle\rangle$ is given as

$$||\rho\rangle\rangle = ||\rho_1\rangle\rangle \oplus .... \oplus ||\rho_n\rangle\rangle = ||\rho'\rangle\rangle \oplus ||\rho''\rangle\rangle . \tag{A2}$$

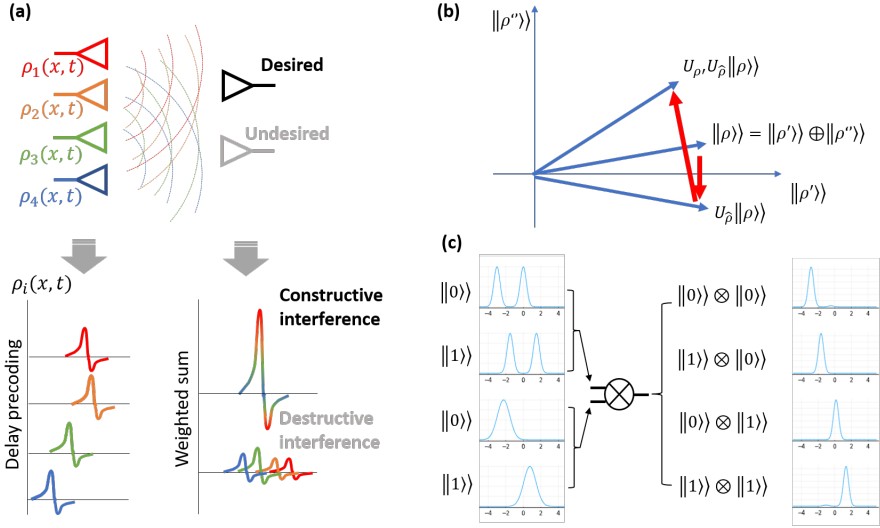

Figure A1: (a) Delay and sum beamforming in wireless communications (freespace is to be replaced with a waveguide network for neural signals); (b) Grover algorithm with cubits; (c) A tensor product state with two cubits. All take advantage of classical wave physics with the Euclidean norm.

This corresponds to the following $m$ qubit state:

$$|\rho\rangle = \rho_1 |0_0\rangle \otimes |0_1\rangle \otimes ... \otimes |0_m\rangle \oplus \rho_2 |1_0\rangle \otimes |0_1\rangle \otimes ... \otimes |0_m\rangle \oplus .... \rho_n |1_0\rangle \otimes |1_1\rangle \otimes ... \otimes |1_m\rangle . \quad (A3)$$

So, $n$ cubits and $m$ qubits span the same state space of $n = 2^m$ dimension. Then the following exactly the same two unitary operations, starting with $||\rho\rangle\rangle = ||\rho^{init}\rangle\rangle$, are repeated $O(n^{1/2})$ times on the cubit state:

$$U_{\rho'} = 2 ||\rho\rangle\rangle \langle\langle\rho|| - I, \quad (A4)$$

and

$$U_{\rho''} = \begin{cases} -||\rho_i\rangle\rangle & \text{if } ||\rho_i\rangle\rangle = ||\rho''\rangle\rangle , \\ ||\rho_i\rangle\rangle & \text{otherwise.} \end{cases} \quad (A5)$$

Though an exponentially larger number of cubits is required (this may make sense considering $\sim 10^{11}$ neurons in our brain), the complications associated with encoding $n$ data into $m$ qubits can be avoided.

In contrast, the exponential speedup in specific QC algorithms, such as quantum Fourier transform and factoring, seems difficult since they take full advantage of tensor product state spaces. Though tensor product state of cubits, such as

$$||\rho_i\rangle\rangle \otimes ||\rho_j\rangle\rangle \quad (A6)$$

can be defined and constructed with signal multipliers as exemplified in Fig. A1(c), the states can entangle only locally and the dimension is limited by the bandwidth.

There are other distinct differences to be mentioned. The interference discussed here of classical waves can occur for signals from different sources. This is a noticeable difference since the qubits interfere only from the same sources. In addition, the coherence time of classical waves can be quite long even at room temperature, as is observed in sound waves, radio waves, ocean waves, and so on [31–34].