# OpenReview forum: "Operator-Discretized Representation for Temporal Neural Networks"
_NeurIPS.cc/2022/Conference — NeurIPS 2022 Submitted_

### Official Review · Reviewer_k9YU · 2022-07-08

**Rating:** 3
**Confidence:** 2
**Soundness:** 2 fair
**Presentation:** 1 poor
**Contribution:** 2 fair

**Summary:**

The authors propose an alternative representation of artificial neural networks through the use of machinery usually used to analyze complex quantum systems.  Through formalisms from operator algebra, they are able to link analog neural networks (ANNs) and spiking neural networks (SNNs) through different choices of operators.   For example, operator representations typically applied to quantum wave packets are ported to artificial neural networks under the formalism outlined by the authors.  Their proposed framework also allows for more advanced detection strategies in comparison to simple threshold detection commonly used.  They also show the practical benefits of their representation.

Past this, they they develop physical representations for the creation and annihilation operators for neural networks through considering transmission line models commonly used in quantum electrodynamics.  Their operator formalism then also allows them to consider the case of non-stationary neural signals, defining appropriate time dependent operators through re-use of operators they defined in the stationary case.

**Questions:**

Could the authors clarify -- when reading the introduction of the paper, the explanation of clock timings and comparing them to membrane time constants makes me think the point trying to be made there was to say that it is not the dynamics following much lower time scales that is important correct? but rather it is the dynamics that follow much larger time scales?  If this is the case, after that analogy what I am failing to see is how the proposed representation takes advantage of this fact; I think it would be extremely helpful to hammer away at the relation throughout the paper since it seems like such an important part of the introduction.

I also think more clarification related to Fig. 2 is required.  I do not think I have the necessary requisite knowledge to fully understand the LC model for signals traveling between axons/dendrites -- but there is a great deal going on in the figure.  I am curious as well, although the LC model has the advantage as you say that it can better transmit information, does this mean it is not possible to create an equivalent RC model?

**Limitations:**

The greatest limitations of the paper in my opinion are the fact that it requires a great deal of requisite knowledge that many will not have.  Fleshing out the appendix here would help a lot — and I think the authors really have to guide the reader a great deal more.  For example, even jumping into the modified bra and ket notation for the newly introduced ‘cubits’ was a bit overwhelming and I think not necessarily well motivated.

**Strengths And Weaknesses:**

The main strength of the paper in my opinion is attempting to come up with a general framework to describe neural computation using notions from quantum computing.  As the authors note, this kind of theory goes both way — i.e. it can aid in certain aspects of quantum computing, but it may also provide another means to understand neural computation.  In addition to that, the authors also try to show the practical consequences/benefits of their framework by showing how it facilitates the conversion from temporally coded SNNs to ANNs.  As hardware implementations of neural computing advance in parallel to quantum computing over time, I believe that drawing these analogies and coming up with a cohesive framework can be of great benefit.


However, I find much of the paper difficult to read both conceptually and theoretically.  In my opinion the paper assumes working knowledge of many different fields/concepts that a great deal of readers would struggle to grasp, i.e. quantum electrodynamics, categorical theories, operator theories, and so on.  I would be okay with this if the references provided at least served as good introductions to these concepts and were presented in a way that I would be at least able to follow high level ideas presented in the paper.  Greatly expanding the appendix in this case would not just be a benefit to the common reader, but a necessity in this case.  I appreciate what the authors are trying to do, but the presentation in my opinion needs a great deal of work.

---

> ### Author Response · Authors · 2022-08-04
> **Could you help us by better clarifying your comments?**
>
> Thank you so much for spending your time on the review.
>
> First of all, we are glad to find that the motivation, framework, and benefit of our paper have been correctly captured on the whole. Also, we appreciate your sincere understanding on our challenge of writing a multidisciplinary paper.
>
> Here we would like to ask for your additional clarification to improve our paper. Frankly, we are a bit lost on how to handle your following comment:
>
> "I would be okay with this if the references provided at least served as good introductions to these concepts and were presented in a way that I would be at least able to follow high level ideas presented in the paper."
>
> "Greatly expanding the appendix in this case would not just be a benefit to the common reader, but a necessity in this case."
>
> We thought we had already provided some excellent textbooks and review papers (such as ref 24-27, 38) in the field to deep dive into each content, and we are planning to add some more (such as P. Dirac's textbook for the original bra/ket notation). However, we do not see much reason to just blindly duplicate the content of those textbooks and reviews in our appendix.
>
> So, could you be more specific about which part you have difficulties with? That kind of specific suggestion should greatly help us to improve the presentation of our paper for the NeurIPS audience.  Otherwise, it may be like asking Einstein to explain his general relativity theory starting with differential geometry from scratch w/o identifying which part is difficult to understand.
>
> For your first question, in short, both time scales are important, but the larger time scale can be handled with other techniques, such as time/positional embedding. Therefore, our focus of the present paper is to show how to systematically handle the smaller time scale. We will clarify this point better in the update.
>
> For your second question, just in case you are not familiar with distributed RC v.s. LC transmission line, please refer to literature on telegrapher equation (e.g., https://en.wikipedia.org/wiki/Telegrapher%27s_equations, lossy transmission line). RC and LC play fundamentally different roles since they correspond to the diffusion equation and the wave equation, respectively.  Operators for RC inherently ask to include dissipative loss via an imaginary part in the frequency. Thus, when forward signals are diminishing, backward could be exploding, which can make the system unstable, similar to the well-known gradient vanishing/explosion issue.
>
> Finally, we consider cubit (we should name it!) should be appropriately defined to bind the discrete spike physical signal to the neural information entity in the state space different to qubits (that is why we use double bra/ket):
>  half or full => whether to include inhibitory
>  normalized or unnormalized =>  algorithmically normalize or not
> If appropriate, we may move this to the appendix for a more complete explanation.
>
> We appreciate your further advice to improve the readability and presentation of our paper for NeurIPs audience.

---

> > ### Comment · Reviewer_k9YU · 2022-08-06
> > **Response**
> >
> > Thank you for the detailed reply.  I saw these references originally, and tried to comb through them however, not having a strong background in this area made it difficult.  While I do not think you should have to blindly duplicate information, I believe it is important that time be spent to communicate these ideas in a way that would maximize the amount of people able to read the paper and walk away with new insights.  For example, even the bra-ket notation that I am slightly familiar with can be slightly confusing since it is not something I see very often;  as another example, equation 16 seems very abrupt and did not bring me many additional insights.  While the working details of general relativity or not simple,  I am sure that there exist a layman explanation that many people can grasp.

---

> > > ### Author Response · Authors · 2022-08-08
> > > **We will work on it.**
> > >
> > > Thank you for your clarification. I see your point better. We will update it with this focus. If you have other points for us to clarify, please let us know ASAP.

---

### Official Review · Reviewer_8i1a · 2022-07-11

**Rating:** 3
**Confidence:** 1
**Soundness:** 1 poor
**Presentation:** 1 poor
**Contribution:** 1 poor

**Summary:**

The authors propose a mathematical framework based on methods in quantum physics and category theory to interpret analog and spiking neural networks. I did not understand the motivation behind the study nor its contribution to the field. I am entering a brief review with low confidence score because the techniques employed in this paper are well outside my area of expertise.

**Questions:**

None.

**Limitations:**

I don't believe there are any negative societal impacts or ethical concerns. My concerns regarding limitations are summarized above.

**Strengths And Weaknesses:**

I will not comment on the technical details of this paper -- I have asked that the area chairs consider replacing me with another reviewer with more domain knowledge in quantum physics.

Nevertheless, I will comment briefly on the appeal of this paper to a general machine learning audience. In short, the paper is not well-written to appeal to this audience and on these grounds alone it may be justified to reject this paper. The abstract and introduction do not explain the basic ideas that will be employed and instead drops hints and jargon to those with a strong physics background ("It is tempting for those with some physics background to apply techniques..." and "operator algebra has been applied to Hopfield networks [18]"... without defining operator algebra.) I expect the technical details of this paper would not be easily digested to the average NeurIPS reader (e.g. Bra–ket notation is used without any explanation).

Perhaps most importantly for a NeurIPS audience, the practical demonstrations of this work feel weak and lacking. The authors only show a brief proof-of-principle on MNIST. If this result is surprising or significant it is lost on me and should be better explained.

Finally, I also want to note that several comments in the paper referring to biological networks strike me as concerning. For instance, "the biological brain operates with low-frequency brain waves closer to our behavioral time scale" -- is a dubious motivation for this study since (a) other neural oscillations are much faster than the theta frequency cited by the authors here, and (b) the biological function of all neural oscillations is still poorly understood and it is particularly controversial to suggest that these oscillations operate like a "clock"

---

> ### Author Response · Authors · 2022-08-03
> **The most important weakness raised is misunderstood**
>
> Thank you for spending your time reviewing the paper.
>
> You mentioned that the most important weakness of our paper is limited demonstration i.e. the paper only contains MNIST demonstration. However, there is a serious misunderstanding. The fat is that our paper also contains results on accuracy and throughput for CIFAR10&100 (with 10 different seeds) as given in section 4.2 as well as fig. 4.

---

### Official Review · Reviewer_DDpm · 2022-07-11

**Rating:** 3
**Confidence:** 2
**Soundness:** 3 good
**Presentation:** 2 fair
**Contribution:** 2 fair

**Summary:**

The authors used the linear algebra which is well established in the quantum theory to rewrite the transmission line model, which is as fine as the cable model. By adding some assumptions, they built the tSNN (operator) model which likes a rate model including the time delay information. The keys to simplify the cable-level model to the rate-level model are the use of the constant velocity assumption which can eliminate the distance x and the use of the inner-product-based detection which can eliminate the time t.

**Questions:**

Overall, to me, this work does not bring any really new contribution to neuroscience and AI (see the above). Please clarify it.

**Limitations:**

The authors should address the limitations of their model in detail, in particular, those assumptions used which are not biologically plausible.

**Strengths And Weaknesses:**

Strengths:
1) The work makes a connection between a cable-level model to rate-level model, though there are some unclear issues (see below).

Weaknesses:
1) From the perspective of neuroscience, from the start, the motivation of setting the global clock T_c closer to our behavioral time scale is wrong. At the circuit level (not the function level), the neural dynamics is continuous. The signal detection process (Eq.16) is far from reality. There is a gap between the aggregated signal (Eq.29) and the event firing time (Eq.30), which needs to clarified by the authors. This gap leads to another question that all neurons in the model share the same time clock (Eq.32), which is not true in reality.
2) From the perspective of artificial intelligence, the author gave a model (Eq.30 or Eq.33) which is equivalent to a feedforward network in some cases. However, it seems that the connection matrix in the equivalent feedforward network have to remain the same between layers. Though the author claims that their tSNN can encode the time delay information, the model (Eq.33) has just two variables which is not interesting in AI. In fact, we can intuitively get the ultimate model (Eq.33) without the complex linear algebra.

---

> ### Author Response · Authors · 2022-08-03
> **Clarifying your question**
>
> Thank you very much for spending time reviewing our paper.
>
> First of all, as is written in the first sentence of the abstract. the paper is on the artificial brain, not on the biological brain. We proposed a new representation and clarified the condition (not a restriction) in Theorem 1 such that tSNNs can more efficiently (energy, etc) process existing ANNs benchmarks/workloads. So it does not matter much whether our formulation reflects 100% of what the biological brain does.  We consider our simpler formulation such as the detection w/o the gap, not a limitation, but instead a desired feature to start with when integrating billions of artificial neurons.
>
> Thus, we consider weakness (1) from a neuroscience perspective to be rather irrelevant. Setting the global clock T_c "closer" to our behavioral time scale is "inspired" by well-known neuroscience literature (place cell, 2014 Nobel prize in our ref 17). which we believe is newly incorporated into tSNNs for the above benefit.
>
> Then, let us move on to weakness (2). You are correctly identifying that our present formulation is missing the layer dependency on w. So this typo has to be updated in the revision. Indeed as demonstrated in the evaluation section, w's in different layers have correctly been taken differently for the benchmarks to work.
>
> Please kindly let us know whether this explanation makes sense or if you still have issues.
>
> We do appreciate your time spent on this matter.

---

> ### Author Response · Authors · 2022-08-05
> **Further clarification**
>
> We think your last statement in weakness 2: "we can intuitively get the ultimate model (Eq.33) without the complex linear algebra." is incorrect. What you mentioned may be true if X_i = (x_i, t_i) i.e., just naively representing spatiotemporal coordinate (let us ignore the layer index (n) for simplicity here),
>
> However,  we derived a specific relationship of  X_i = x_i f(t_i) as written in line 180 with f(t_i) in Eq. 27. This specific choice cannot be obtained without incorporating the actual spike physical dynamics discussed earlier in sections 3.1-3.4.

---

### Official Review · Reviewer_p6e7 · 2022-07-17

**Rating:** 4
**Confidence:** 2
**Soundness:** 2 fair
**Presentation:** 2 fair
**Contribution:** 2 fair

**Summary:**

The paper applies a formalism from quantum theory to ANNs and SNNs that offers to deal with time discrete and asynchronous events. The theory is applied to temporal spiking neuronal networks (tSNNs) that are shown to be equivalent under certain conditions to ANNs. The coding is applied to CIFAR 10 & 100, and it is shown that the suggested operator-discretized tSNN have a better “throughput” than the Euler. discretization of tSNN, and that operator-discretized tSNN are comparable in “throughput” with ANN.

**Questions:**

(1)	The equivalence between tSNNs and ANNs is only shown with some restrictions that are not so clearly spelled out. In general, tSNNs are treated as ANNs in the literature mainly for time-to-first-spike (TTFS) coding, assuming an overall clock that resets the state of the system at discrete times, and from there on considers time as the analog variable of the ANN. This special case does not include the processing of multiple spiking signals, emitted from each single neuron, in real and continuous time, but never synchronized with an external clock or with other neurons of the network. Does you theorem contribute to this more general case? In what sense does it go beyond the well studied TTFS coding?

(2)	Event-based simulations of spiking neurons has a long tradition and is shown in many papers and simulation platforms to dramatically outperform naïve Euler-discretized simulations of spiking networks. In what sense does your approach go beyond these event-based simulations of SNNs?

(3)	If the aim is to contribute to the simulation of tSNNs, then the comparison in the simulations should be done with state-of-the art methods to simulate tSNNs. Can you show such results?

(4)	So far the paper simply shows that the performance is comparable to ANNs, supporting the theorem, but begging the question whether the technique can really be used for the analysis of SNN on behavioral time scales, as explained in the Introduction. It is said that at present, the networks are not fully asynchronous, and this is justified by the slow waves in the brain. But slow waves are seen on the level of field potentials, and are far away from globally imposing spike times in individual neurons, as it is assumed in the paper. Can you give a hint how the technique will overcome this?


**Limitations:**

yes

**Strengths And Weaknesses:**

It is certainly always interesting to transfer successful methods from other fields. Yet, in the current example, it does not become clear, what one really gains over existing techniques in simulating ANNs, or tSNNs transformed to ANNs.

---

> ### Author Response · Authors · 2022-08-03
> **Clarifying your questions**
>
> Thank you very much for spending time reviewing our paper. Please let us confirm your points to facilitate our understanding of your questions and the overall review process:
>
> (1) One of our main motivations is to clarify the condition (not a restriction) such that tSNN can become equivalent to ANN as proven in theorem 1 via our new representation. Furthermore, we actually try to overcome the restrictions of TTFS you pointed out  (no multiple spikes in a single overall global clock period, limited temporal correlation/control) and go beyond. Indeed, our model does not have those restrictions, thanks to the wave-based superposable signal model in an "analogy" to the boson model as written in lines 105-6, and more complex delta_t correlations, for example, via Eqs. 27.
>
> We are planning to update our paper to include specific comparisons to TTFS. Does this sounds reasonable or do we miss any of your points?
>
> (2-3)  Could you clarify specifically whether you are requesting to augment the descriptions on already referred literature (lines 26-9) or to include a new one we might be missing?  In the latter case,  could you navigate us to the specific state-of-the-art literature you have in mind to facilitate the review process? We are aware that even-based simulation has a long tradition as some of the prior-art works and their limitations have been briefly described in lines 26-9. However, we have so far not yet found any event-driven simulation literature for tSNN that has proven to run standard benchmarks competitively (both throughput and accuracy) to ANNs.
>
> (4) The use of a slow global clock in the biological brain should be well known in the neuroscience literature, for example, in ref 17 (place cell) since the discovery won the Nobel prize in 2014. So we consider that the hint you asked for is already there. On the other hand, as you pointed out, the detailed microscopic mechanisms of the biological brain, such as how the field potential controls individual neurons, may not have been fully understood. However, they should not be considered as a serious show stopper when building an artificial brain, as the detailed biological mechanism of how the birds fly was irrelevant to building airplanes. So, we consider that your question has been already answered to an appropriate level. Could you clarify your question further in case you disagree?
>
> We do appreciate the further time you send on this.

---

> > ### Comment · Reviewer_p6e7 · 2022-08-07
> > **I simply fear you reproduce in another language what has already been done.**
> >
> > Ok, so you simply consider a repetitive application of TTFS networks for each period of some theta-oscillation (say), while completely resetting the neurons with each period? If so, this would be a trivial extension. You seem to go beyond when you say you consider multiple spike interactions within a period. It is not clear, how you map this to ANNs, and what the role of the period is. Is the point that you consider only a finite number of spikes within a period, and the size of the ANN you map the tSNN depends on the number of spikes?
> >
> > If there is no upper bound on the number of spikes within a period (I don’t see such an upper bound in your proposals), then you seem to simply describe event-based implementation of spiking neural networks, as most spiking network simulators do it, see e.g. the NEST simulator that is itself based on many papers. In this case you should show that your simulations are faster than those classical event-based simulations.
> >
> > Since you stress the global clock so much, there must be something in between time-to-fist-spike and event-based spike simulations. You should be able to describe this “more” in a language does not use your particle physics. Can you do that? If not, I fear you are just replicating in another language what others did (but I’m happy to be taught differently).

---

> > > ### Author Response · Authors · 2022-08-08
> > > **This is a paper on a new scalable representation to facilitate temporal neural network computing**
> > >
> > > First of all, as the title says, our paper is on a new representation (or language in your wording) for temporal neural networks. We do not have any objection in claiming that our representation covers tSNN/TTFS in a certain condition (like proposition 2) and event-driven approaches for more unrestricted x_i in Eq. 30.
> > >
> > > Then the bottom-line question is what for? Presently it is to prove out scalable performance competitive to the mainstream ANNs, which we believe is an essential condition for spike-based (or more generally even-driven) systems to remain a viable option for future AI systems. We have so far found no midsize benchmark results in the literature, such as CIFAR10/100 or higher as we demonstrated in sec 4.2. We only found various MNIST results for tSNN/NEST in the literature. For example, tSNN/TTFS for our ef. 12  reproduced in Figure 3 as tSNN(Euler), or NEST in the paper entitled Evaluation of the Effect of the Dynamic Behavior and Topology Co-Learning of Neurons and Synapses on the Small-Sample Learning Ability of Spiking Neural Network Xu Yang * , Yunlin Lei, Mengxing Wang, Jian Cai, Miao Wang , Ziyi Huan and Xialv Lin, file:///C:/Users/084566760/Downloads/Evaluation_of_the_Effect_of_the_Dynamic_Behavior_a.pdf.
> > > Those MNIST results are behind  ANNs in accuracy and/or throughput.
> > >
> > > Once we validate our approach for existing models and workloads, it should be quite interesting to investigate new functionalities maybe in between tSNN/TTFS and event-driven approaches as you pointed out. We should go step by step cautiously since it may not be a good practice to build a house on shaky ground. Since intelligence only appears in an above-threshold number of neurons, a scalable representation is MUST.  Our intention is to present our new scalable representation to facilitate this kind of research in the community.
> > >
> > > We come up with this new scalable representation by exploiting physics language. As discussed in our appendix, other methods such as kernel methods might do similar for the logical layer, but may not work as well when combined with the physical layer.  The reason why we decided to adopt the more physics-oriented representation such as with spike creation/annihilation operators is to provide us a representation more transparent to spike physical dynamics, such as X_i = x_i f(t_i)  in line 180 for the specific f in Eq. 27.
> > >
> > > Hope you see our point.

---

### Meta-Review · Area_Chair_kjSN · 2022-08-23

**Recommendation:** Reject
**Confidence:** Certain

**Metareview:**

Reviewers agree that manuscript presents a fresh attempt, but also that the manuscript is lacking in several aspects. The writing has a lot of room for improvement and not suitable for the NeurIPS community. It's neuroscientific claims are controversial, or relies on non-mainstream arguments without appropriate justifications. The theoretical and experimental results are limited.

**Award:**

No

---

### Decision · Program_Chairs · 2022-09-14

Reject